# COVID-Net USPro: An Explainable Few-Shot Deep Prototypical Network for COVID-19 Screening Using Point-of-Care Ultrasound

**DOI:** 10.3390/s23052621

**Published:** 2023-02-27

**Authors:** Jessy Song, Ashkan Ebadi, Adrian Florea, Pengcheng Xi, Stéphane Tremblay, Alexander Wong

**Affiliations:** 1Department of Systems Design Engineering, University of Waterloo, Waterloo, ON N2L 3G1, Canada; 2Digital Technologies Research Centre, National Research Council Canada, Toronto, ON M5T 3J1, Canada; 3Department of Emergency Medicine, McGill University, Montreal, QC H4A 3J1, Canada; 4Digital Technologies Research Centre, National Research Council Canada, Ottawa, ON K1A 0R6, Canada; 5Waterloo Artificial Intelligence Institute, Waterloo, ON N2L 3G1, Canada

**Keywords:** ultrasonic imaging, lung, COVID-19, few-shot learning, deep explainable architecture

## Abstract

As the Coronavirus Disease 2019 (COVID-19) continues to impact many aspects of life and the global healthcare systems, the adoption of rapid and effective screening methods to prevent the further spread of the virus and lessen the burden on healthcare providers is a necessity. As a cheap and widely accessible medical image modality, point-of-care ultrasound (POCUS) imaging allows radiologists to identify symptoms and assess severity through visual inspection of the chest ultrasound images. Combined with the recent advancements in computer science, applications of deep learning techniques in medical image analysis have shown promising results, demonstrating that artificial intelligence-based solutions can accelerate the diagnosis of COVID-19 and lower the burden on healthcare professionals. However, the lack of large, well annotated datasets poses a challenge in developing effective deep neural networks, especially in the case of rare diseases and new pandemics. To address this issue, we present COVID-Net USPro, an explainable few-shot deep prototypical network that is designed to detect COVID-19 cases from very few ultrasound images. Through intensive quantitative and qualitative assessments, the network not only demonstrates high performance in identifying COVID-19 positive cases, using an explainability component, but it is also shown that the network makes decisions based on the actual representative patterns of the disease. Specifically, COVID-Net USPro achieves 99.55% overall accuracy, 99.93% recall, and 99.83% precision for COVID-19-positive cases when trained with only five shots. In addition to the quantitative performance assessment, our contributing clinician with extensive experience in POCUS interpretation verified the analytic pipeline and results, ensuring that the network’s decisions are based on clinically relevant image patterns integral to COVID-19 diagnosis. We believe that network explainability and clinical validation are integral components for the successful adoption of deep learning in the medical field. As part of the COVID-Net initiative, and to promote reproducibility and foster further innovation, the network is open-sourced and available to the public.

## 1. Introduction

The Coronavirus Disease 2019, or COVID-19, caused by severe acute respiratory syndrome coronavirus 2 (SARS-CoV-2), has been continuously impacting individuals’ wellbeing and the global healthcare systems [1]. Despite the vaccination efforts, policies, and regulations in place, due to the rapid transmission of the virus and waves of rising cases, the development of effective screening and risk stratification methods remains to be a critical need in controlling the disease [2]. Various types of diagnostic tools, including reverse transcription-polymerase chain reaction (RT-PCR), rapid antigen detection tests, and antibody tests, have been developed and adapted globally to increase the rate of screening. While RT-PCR has been the gold standard test for diagnosing COVID-19, the technique involves large labour and laboratory resources and is time-consuming [3]. Other rapid antigen tests and antibody tests with varying sensitivity are also less reliable in comparison to RT-PCR tests [3].

For people with significant respiratory symptoms, medical imaging is used to identify the disease and assess the severity of the disease progression [4]. Under this protocol, a computed tomography (CT) scan, chest X-ray (CXR), or point-of-care ultrasound (POCUS) imaging can be performed and used clinically as an alternative diagnostic tool [2]. To make a diagnosis, acute care physicians and radiologists visually inspect the radiographic images to find patterns related to symptoms and to assess the severity of COVID-19 infection and deformation [3]. During times of high transmission rate of COVID-19, the large influx of patients increases the burden on clinicians and radiologists. Medical image processing and artificial intelligence (AI) can assist in reducing this burden and accelerate the diagnostic and decision-making process, as existing models and algorithms continue to improve and the amount of available medical image data continues to grow [5,6,7].

Different imaging modalities, including CT scan, X-ray, and ultrasound, may be used in the diagnosis of COVID-19 and offer varying diagnostic values [8]. Chest CT scan is the most sensitive imaging modality in the initial diagnosis and management of confirmed cases, but it is more expensive and time-consuming [5,8]. In contrast, ultrasound imaging is more accessible and portable, cheap, and safer, as radiation is not involved during the examination, which are desirable properties for its usage, especially in resource-limited settings/environments/areas/regions [8].

Deep learning usually requires a large set of training examples [4,7,9]. However, due to the nature of novel diseases, the availability of such a huge amount of well annotated data poses a great challenge to learning algorithms. Few-shot learning is an approach where a model is trained to classify new data based on a limited number of samples exposed in training [10]. This approach resembles how humans learn, as we can recognize new object classes from very few instances, and it is different from conventional deep neural networks that require a large amount of data in the training phase [10]. Since the few-shot model requires much fewer data to train, the computational costs are also significantly reduced [10]. These properties make it an appropriate and promising approach for COVID-19 as it relates to novel and rare disease diagnosis. One approach for few-shot learning is metric-based learning. As a few-shot metric-based learning approach, prototypical networks (PN) perform classification by computing distances to prototype representations of each class [10]. PN has shown state-of-the-art (SOTA) results on other datasets and domains (e.g., [11,12,13]), proving that simple design decisions can yield significant improvements over other complicated architectures and meta-learning approaches [10].

In this work, we present an open-source explainable deep prototypical network, called COVID-Net USPro, that can detect COVID-19 cases with high accuracy, precision, and recall from a very limited number of lung ultrasound (LUS) images. When trained with only 5 shots, COVID-Net USPro classifies between positive and negative COVID-19 cases with 99.55% overall accuracy, 99.93% recall, and 99.83% precision for COVID-19 positive cases. Intensive experimentation was conducted (e.g., testing different image encoders, varying training conditions, and the number of classes to optimize the network) to assess the performance of COVID-Net USPro. To ensure the network’s fairness and accountability, an explainability module is constructed to assess the network’s decisions with visual explanation tools, i.e., Grad-CAM [14] and GSInquire [15]. Moreover, our contributing clinician (A.F.) carefully verified and validated the pipeline, as well as the results, to ensure the validity of the proposed solution from the clinical perspective. To facilitate the adoption and openness of AI in healthcare, support reproducibility, and encourage innovation, the network, and all the experiment scripts, are open-sourced at the project’s Github repository.

Our work contributes to the existing body of literature on rare disease medical image analysis using few-shot learning in at least the three following ways:We present a few-shot network that reaches 99.55% accuracy when trained with only five shots, while other related works achieving similar (or lower) performance require larger numbers of training examples (see details in Section 2).COVID-Net USPro is an explainable network, as demonstrated quantitatively by analysis from two explainability visualization tools and qualitatively by our clinician validation.COVID-Net USPro is open-sourced and available to the public, which helps promote the reproducibility and accessibility of AI in healthcare. This would encourage further innovation in the field of deep learning applied to medical image analysis for novel disease diagnosis.

The remainder of this paper is structured as follows. Section 2 describes related previous studies and highlights their limitations. Section 3 explains data, techniques, and the experiments conducted to assess the network performance in detail. Section 4 presents findings of the research, including quantitative performance assessment and qualitative explainability analysis results. Results are then discussed and summarized in Section 5, where some limitations of the research and future directions are also presented.

## 2. Related Work

There are several studies that aim to apply deep learning to the screening and detection of COVID-19 cases. As an open-source and open-access initiative, the COVID-Net [5,7,9,16] includes research on the application of deep learning neural networks using a multitude of image modalities, such as CT, X-ray, and ultrasound images. Multiple works have demonstrated the effectiveness of deep learning in the classification of CT and X-ray images. For example, Aboutalebi et al. proposed the COVID-Net CXR network [17], which is a tailored deep convolutional neural network (DCNN/CNN) for the detection of COVID-19 cases. The network that was trained on 5210 chest X-ray images achieved an overall accuracy of 98.3% and 97.5% sensitivity for COVID-19 cases. In another work, Ozturk et al. [18] proposed a DCNN network that was previously used for the you-only-look-once (YOLO) real-time object detection system to classify X-ray images. Their proposed network achieved 98.08% accuracy for binary COVID-19 case detection. Afshar et al. [19] proposed a capsule network, called COVID-CAPS, which achieved over 98% accuracy and, specificity, trained on 112,120 X-ray images. Gunraj et al. [6] proposed the COVID-Net CT network for COVID-19 detection from CT images. Being trained on 194,922 CT images, the network scored 96.2% in sensitivity and 99% in specificity for COVID-19 cases. The potential of including both CT-scan and X-ray images for classification is also explored. For instance, Thakur [20] presents a DCNN-based model achieving over 99% accuracy and precision for COVID-19 detection using 11,095 X-ray and CT images. Few works proposed DL-based solutions for COVID-19 detection from ultrasound images. In a recent study, MacLean et al. [7] proposed a highly efficient self-attention deep neural network, called COVID-Net US, for COVID-19 detection from POCUS ultrasound imaging. The network was trained on 3947 positive COVID-19 and 3697 negative normal case images and achieved an area under the receiver operating curve (AUC) of over 98%. In another study, Diaz-Escobar et al. [21] leveraged pre-trained neural networks, such as VGG19 [22], InceptionV3 [23], and ResNet50 [24], in the detection of COVID-19 from ultrasound images and achieved 89.1% accuracy and AUC of 97.1%. One main requirement of conventional deep neural network architectures in most of the existing research is a large amount of training data, as in all the mentioned works above, datasets surpassed 5000 total images [7,9,17]. This may pose a serious limitation in the case of novel/rare diseases and new pandemics where not many images are available for training the DL model.

The application of few-shot learning techniques has also been investigated. For example, Shorfuzzaman et al. [25] proposed MetaCOVID, a Siamese few-shot neural network with the contrastive loss for detecting COVID-19 using CXR images. The performance of the best network achieved an accuracy of 95.6% and an AUC of 97% when tested in a 10-shot setting in the inference phase. In another work by Ebadi et al. [26], a deep Siamese convolutional network called COVID-Net FewSE was proposed that can detect COVID-19 positive cases with 90% recall and accuracy of 99.93% when the network is provided with only 50 examples in the training phase. In the work by Karnes et al. [27], the use of adaptive few-shot learning for ultrasound COVID-19 detection was studied.

Although the feasibility of adopting few-shot learning techniques for COVID-19 detection from medical imaging has been already investigated, most of the previous studies only focused on the performance of the network, analyzing its accuracy, precision, and recall. A comprehensive quantitative analysis of network explainability to ensure that decisions are made based on actual patterns is either missing or inadequate. In addition, a thorough validation assessment, performed by clinicians as the domain experts, is also lacking. There are limited discussions on whether the data interpretation process of these well performing networks aligns with real clinical settings [25]. These limitations could jeopardize the full understanding of the network and hinders the adoption of the network in the real clinical environment. In this work, we aim to address these limitations by presenting a high-performing network that can accurately detect COVID-19 cases when presented with only five shots. We also perform a comprehensive explainability analysis to validate network behaviour. Additionally, most importantly, we further validate the network and findings by an experienced practicing clinician.

## 3. Materials and Methods

### 3.1. Data

We use the COVIDx-US dataset v1.4. [1] as the data source. COVIDx-US is an open-access benchmark dataset of lung ultrasound imaging data that contains 242 videos and 29,651 processed images of patients with COVID-19 infection, non-COVID-19 infection (mainly pneumonia), other lung conditions, and normal control cases. The dataset provides LUS images captured with two kinds of probes, linear probes, which produce a square or rectangular image, and convex probes, which allow for a wider field of view [28]. Due to the difference in the field of view, combining the linear and convex probe data in training may increase noise and influence the performance of the network. As there are also a low number of COVID-19 positive examples captured with the linear probes in the dataset, we exclude them from this study. A total number of 25,262 convex LUS images are then randomly split into the training set, containing 90% of images, and the unseen test set with the remaining 10% of images, ensuring all frames from each video are either in train or test set to avoid data leakage. The training set is then split into 80–20%, representing the training and validation datasets. The validation dataset is used for hyperparameter tuning and performance assessment in the training phase. All images are re-scaled to 224×224 pixels to keep the images across the entire dataset consistent. The dataset is further augmented by rotating each image by 90°, 180°, and 270°, resulting in a total of 101,048 images (25,262×4). This rotation technique is an appropriate method for increasing the dataset size, as it keeps the images and areas of interest for clinical decisions unaltered and in-bound [29].

### 3.2. Methods

COVID-Net USPro is a prototypical few-shot learning network that trains in an episodic learning setting. It uses a distance metric for assessing similarities between a set of unlabelled data, i.e., query set, and labelled data, i.e., support set. Labelled data can be used to compute a single *prototype* representation of the class, and unlabelled data are assigned to the class of the prototype they are closest to. A prototypical network [10] is based on the idea that there exists an embedding in which points in a class cluster around a single prototype representation for the class. During the training phase, a neural network is used to learn the non-linear mapping of the inputs to an embedding space, and a class prototype is computed as the mean of its support set data in the embedding space. Classification is then done by finding the nearest class prototype for each query point based on a specified distance metric. An episodic approach is used to train the model, where in each training episode, the few-shot task is simulated by sampling the data point in mini-batches to make the training process consistent with the testing environment. The performance of the network is evaluated using the unseen test set. Both quantitative performance analysis based on accuracy, precision, and recall and qualitative explainability analysis are conducted. A high-level conceptual flow of the analysis is presented in Figure 1.

We define the classification problem as a *K*-way *N*-shot episodic task, where *K* denotes the number of classes present in the dataset, and *N* denotes the number of images for each class in each episode. For a given dataset, *N* images from each of the *K* classes are sampled to form the support set, and another *M* images from each class are sampled to form the query set. The network then aims to classify the images of the query set based on the K∗N total images presented in the support set. In this work, we consider three classification scenarios and formulate the problem as 2-way, 3-way, and 4-way classification problems. Details are included under Section 3.3.3.

The few-shot classification with a prototypical network can be summarized into three steps: (1) encoding of the images, (2) generating class prototypes, and (3) assigning labels to query samples based on distances to the class prototypes. Let $S=\{\left(x_{\left(1,s\right)},y_{\left(1,s\right)}\right),\ldots ,\left(x_{\left(N,s\right)},y_{\left(N,s\right)}\right)\}$ and $Q=\{\left(x_{\left(1,q\right)},y_{\left(1,q\right)}\right),\ldots ,\left(x_{\left(N,q\right)},y_{\left(N,q\right)}\right)\}$ be the support and query sets, respectively, where each $x_{i}\in R^{D}$ is a *D*-dimensional example feature vector and $y_{i}\in \left\{1,\ldots K\right\}$ is the label of the example. The prototypical network contains an image encoder $f_{\phi }:R^{D}\to R^{H}$ that transforms each image $x_{i}$ onto a *H*-dimensional embedding space where images of the same class cluster together. Class prototypes are then generated for each class by averaging the embedding image vectors in the support set, where $v_{k}=\frac{1}{N}\sum_{i=1}^{N}f_{\phi }\left({x_{i,s}}^{\left(k\right)}\right)$ denotes the prototype of class *k* [10]. To classify the query image, a distance metric is used where distances between the embedding vector of a query image and each of the class prototypes are computed. In this work, the squared Euclidean distance $d\left(v,q\right)=\left||v-q|\right|=\sqrt{\sum {\left(v_{i}-q\right)}^{2}}$ is used, where *q* is the embedding vector of the query image, and $v_{i}$ is the embedding vector of the *i*-th prototype. The choice of the squared Euclidean distance instead of other distance metrics, e.g., cosine distance, is validated by Snell et al. [10], who proved that metrics that are Bregman divergences, e.g., euclidean distance metrics, perform better in the calculation of class prototypes based on embeddings in prototypical networks. After distances are computed, a SoftMax function is applied over the distances to the prototypes to compute the probabilities of the query image being in each class. The class with the highest probability is then assigned to the query image.

In the training phase, the network learns by minimizing a loss function, i.e., the negative log-SoftMax function ($J=-\log\left(p\left(y=k|x\right)\right)$) of the true class *k*. An Adam optimizer with an initial learning rate of 0.001 is used, and the learning rate is reduced if the validation loss is not improved after 3 epochs. In each episode, a subset of data points is randomly selected, forming a support and query set. The loss terms on training and validation sets are calculated at the end of each training episode. To facilitate an effective training process and prevent over-fitting, early stopping is implemented to stop the training process when the validation loss is not improved after 5 epochs. A total of 10 epochs is set for all training processes, and 200 episodes are set for each training epoch. Figure 2 presents an architecture design overview of the COVID-Net USPro network.

The trained model’s performance is evaluated quantitatively and qualitatively. In the quantitative analysis, the model’s accuracy, precision, and recall for each class are analyzed and reported. In the qualitative analysis, model explainability is investigated and visualized. Explainable artificial intelligence (XAI) has been an important criterion when assessing whether neural networks can be applied to clinical settings [30]. While AI-driven systems may show high accuracy and precision in analyzing medical images, lack of reasonable explainability will spark criticism of the network’s adoption [30]. COVID-Net USPro’s explainability is assessed using two approaches, i.e., gradient-weighted class activation map (Grad-CAM) [14] and GSInquire [15], on a selected dataset containing correctly classified COVID-19 and normal cases with high confidence (i.e., >99.9% probability), as well as falsely predicted COVID-19 and normal cases. Grad-CAM generates a visual explanation of the input image using the gradient information flowing into the last convolutional layer of the convolutional neural network (CNN) encoder and assigns importance values to each neuron for making a classification decision [14]. The output is a heatmap-overlayed image that shows the image regions that impact the particular classification decision made by the network [14]. The other tool, GSInquire, identifies the critical factors in an input image that are shown to be integral to the decisions made by the network in a generative synthesis approach [15]. The result is an annotated image highlighting the critical region, which could drastically change the classification result if removed [15]. Results from both tools are reviewed by our contributing clinician (A.F.) with experience in ultrasound image analysis to assess whether clinically important patterns are captured by the network.

### 3.3. Experiment Settings

We comprehensively assess the performance of COVID-Net USPro in detecting COVID-19 cases from ultrasound images by testing various training conditions, such as different image encoders, the number of shots available for training, and classification task types. Details are further discussed in this section.

#### 3.3.1. Image Encoders

To leverage the power of transfer learning, we experiment with multiple encoders, including, but not limited to, the ResNet, DenseNet, and VGG networks [22,24,31]. Pre-trained models refer to using model parameters pre-trained on ImageNet [32]. To concisely summarize the main results, we report the top-4 performing encoders with respect to our research objectives:**ResNet18L1:** Pre-trained ResNet18 [24], with trainable parameters on the final connected layer and setting out features as the number of classes. We consider this pre-trained network as the baseline model for encoders, as it contains the least number of layers and retrained parameters.**ResNet18L5:** Pre-trained ResNet18 [24], with trainable parameters on the last four convolutional layers and final connected layer. Out features set to the number of classes.**ResNet50L1:** Pre-trained ResNet50 [24], with trainable parameters on the final connected layer and setting out features as the number of classes.**ResNet50L4:** Pre-trained ResNet50 [24], with trainable parameters on the last three convolutional layers and final connected layer. Out features set to the number of classes.

#### 3.3.2. Number of Training Shots

The optimal number of shots for maximized performance is tested by training the model under 5, 10, 20, 30, 40, 50, 75, and 100-shot scenarios. For selected models showing a steady increase of performance over increasing shots, 150 and 200-shot conditions are also tested to further verify that the maximum performance is reached at 100-shot. To ensure the training process is faithful to the testing environment, the number of shots for each class presented in each episode is the same in the support and query sets in both training and test phases. For example, in the 5-shot scenario, five images in each class are presented for both the support set and the query set in the training phase, and the same follows in the test phase.

#### 3.3.3. Problem Formulation

In comparison to other classes, e.g., non-COVID-19 and normal cases, the ability of the model to correctly identify COVID-19-positive cases is valued the most. The classification problem for identifying COVID-19 is formulated in three different scenarios as follows, in ascending order of data complexity:**2-way classification:** Data from all three other classes, namely, the ’normal’ class, ’non-COVID-19’ class, and ’other’ class, are viewed as a combined COVID-19 negative class. The network learns from COVID-19 positive and COVID-19 negative datasets in this setting.**3-way classification:** As the ’other’ class contains data from multiple different lung conditions, it has the highest variations and may disrupt the network’s learning process due to the lack of uniformity in the data. In the 3-class classification, the ‘other’ class is excluded, and the network is trained to classify the remaining three classes.**4-way classification:** As the dataset contains four classes, the four-class classification condition remains in this setting, and the network is trained to classify ’COVID-19’, ’normal’, ’non-COVID-19’, and ’other’ classes.

The network hyperparameters and training settings are listed in Appendix A.

## 4. Results

This section summarizes the quantitative performance results of all combinations of experiment settings listed in Section 3.3, as well as the results of the network explaina- bility analysis.

### 4.1. Quantitative Performance Analysis

The performance of COVID-Net USPro is evaluated using the overall accuracy, precision, and recall for each class. As the performance of the model to detect COVID-19 cases is the most important for current clinical use cases, the following report contains precision and recall for the COVID-19 class only. To reduce table size, Table 1 only summarizes the performance of the network under 5-shot and 100-shot scenarios for encoders that scored over 80% across all evaluated metrics. The complete experiment results for all classes, encoders, and shot numbers are detailed in Appendix B.

Across all classification scenarios and models, performance is higher under the 100-shot training condition than in the 5-shot condition, with performance metrics increasing from 5-shot and plateauing after 75-shot, as shown in Figure 3. The ResNet-based networks demonstrate the ability to classify COVID-19 with precision and recall above 87% consistently under both 5-shot, and above 99% under 100-shot conditions. As seen in Table 1, increasing the number of classes (in 3-way and 4-way classification scenarios) reduces the performance of the network. This is expected as the classification problem becomes more complex with more numbers of classes. However, this performance difference among the three classification scenarios is reduced when the number of shots increases, as more examples available for training improve the network’s ability to distinguish between multiple classes. As shown in Figure 3, the performance of ResNet18L1, which has fewer layers than ResNet50L4, is lower than ResNet50L4 at early training. ResNet50L4 is expected to adapt better to input images in early training epochs and extract deeper and more complex representations. In addition, models with more fine-tuned final convolutional layers (i.e., ResNet18L5 and ResNet50L4) achieve higher accuracy, precision, and recall. Therefore, it can be said that, while using pre-trained parameters and simpler models, reducing the computational complexity and space, tailoring parameters on the final convolutional layers to the ultrasound images, and deepening image encoding can boost performance.

In the 2-way and 3-way classification scenarios, it is observed that the precision and recall for classes other than COVID-19 are similar to the COVID-19 class. In the 4-way classification scenario, the precision and recall for the ‘other’ class are ≈2–3% lower than those for ‘non-COVID-19’, ‘normal’, and ‘COVID-19’ classes. This is expected since the ‘other’ class covers various lung conditions/diseases that encompass a larger range of image features and variations. Overall, with precision and recall achieving similar magnitude in the 2-way, 3-way, and 4-way classifications, the network also demonstrates the ability to distinguish between multiple classes/diseases. In comparison to studies outlined in Section 2, the performance of COVID-Net USPro networks, which are tailored to ultrasound images with re-trained parameters, is improved. Accuracy of ResNet50L1 and ResNet50L4 exceeds 98% under a 4-way 5-shot setting while past works such as the MetaCOVID proposed by Shorfuzzaman et al. [25], which also applied a few-shot approach, achieved 95.6% accuracy under a 3-way, 10-shot setting. Additionally, the sensitivity of COVID-Net USPro for COVID-19 cases is also higher than previously reported deep networks that were trained on many images of other modalities such as X-ray or CT, where they scored 97.5% in the best-performing case [6].

### 4.2. Clinical Validation and Network Explainability Analysis

In addition to the intensive quantitative performance analysis, we clinically validated the network outputs to ensure that the network captures important patterns in the ultrasound images. For this purpose, our contributing clinician (A.F.) reviewed a randomly selected set of images and reported his findings and observations. Our contributing clinician (A.F.) is an Assistant Professor in the Department of Emergency Medicine and the ultrasound co-director for undergraduate medical students at McGill University. He is practicing Emergency Medicine full-time at Saint Mary’s Hospital in Montreal.

Figure 4 presents two examples of COVID-19 positive ultrasound images, annotated by Grad-CAM and GSInquire. As seen, the annotated images contain the lung pleura region at the top of the image, while the second example (Figure 4b) also marks the bottom region with high importance. B-lines, or the light comet-tail artifacts extending from pleura to the bottom of the image, and the presence of dark regions interspacing the B-lines at the bottom part of the image, correspond to signs of lung consolidation and are indicators of abnormality [33]. Hence, the visual annotations for the second example (Figure 4b) are more representative of disease-related patterns within the ultrasound image. Figure 4a is one of the examples where the model considers the rib as a structure of interest, which is not an abnormality. Hence, although the model correctly classified the specific image, the decision was made based on invalid clinical factors.

We experimented with two strategies to solve the above-mentioned issue. First, as the current dataset includes images of different qualities, we excluded images with low quality to evaluate their impact on explainability. These low-quality images refer to those with insufficient image depth or the lack of representative features. A severity grade was introduced by COVIDx-US dataset v1.4, called lung ultrasound score (LUSS), which rates each ultrasound video on a scale of 0 to 3 [1]. A score of 0 corresponds to the presence of only normal features, and 3 corresponds to the presence of severe disease artifacts [1]. In the first attempt to improve the network further, images from videos with a score of 0, representing the normal class, and images from videos with scores of 2 and 3, representing the COVID-19 class, are used to train a binary classification model. By observing the annotated images, the network shows to focus more on the bottom regions of the images, though cases, where the network focuses on the top pleura region, are still present. The second strategy to further improve model explainability is to exclude regions above the pleura (i.e., soft tissue) of the images so that the network focuses on the disease-defining features, which are mostly present at the bottom of the images below the lung pleura. Our experiments confirm the effectiveness of this strategy, as the network shows to focus mainly on the bottom regions of the images. Hence, combining the first and second strategies, a binary model with LUSS score filtered and cropped images is trained. Figure 5 shows examples from the analysis after cropping and filtering images. As suggested from the annotated examples and confirmed by our contributing clinician (A.F.), clinically relevant artifacts such as B-lines and lung consolidation are clearly identified in COVID-19-positive images by COVID-Net USPro after implementing the two strategies.

## 5. Discussion

Deep neural network architectures have shown promising results in a wide range of tasks, including predictive and diagnostic tasks. However, such networks require a massive amount of labelled data to train, which is against the nature of new pandemics and novel diseases where there are no or very few data samples available, especially in the initial stages. Using a diverse complex benchmark dataset, i.e., COVIDx-US, we introduced the COVID-Net USPro network, tailored to detect COVID-19 infection with high accuracy from very few ultrasound images. The proposed deep prototypical network leverages deep pre-trained models with fine-tuned parameters on final layers to reduce computational complexity and achieve high classification performance when only 5 examples from each class are presented during the training phase. Accuracy, precision, and recall for the best performing network are over 99%, which are comparable to or outperforming other existing works, even those that used large-scale datasets to train the models [7,27]. As mentioned in Section 2, MetaCOVID network, proposed by Shorfuzzaman et al. [25], which adapted the few-shot learning approach achieved 95.6% accuracy when tested under a 3-way 10-shot condition. COVID-Net FewSE, proposed by Ebadi et al. [26], achieved over 99% accuracy when trained under a 3-way 50-shot setting. In comparison to these networks, COVID-Net USPro is able to achieve over 98.5% accuracy in 2-way, 3-way, and 4-way classification problem settings by using only 5 shots for training. These properties are not only highly crucial for the control of the COVID-19 pandemic, but also for screening patients for new/rare diseases or pandemics as the proposed network can be easily tuned.

In contrast to the lack of interpretability analysis in the previous works, we also intensively assessed the explainability of the network and clinically validated its performance. Experimental results demonstrate that COVID-Net USPro can not only achieve high performance in terms of accuracy, precision, and recall, but also shows predictive behaviour that is consistent with clinical interpretation, as validated by our contributing clinician (A.F.). In addition, as part of the explainability-driven performance validation process, we proposed and implemented two strategies to further improve the network performance in accordance with the background clinical knowledge in identifying COVID-19 positive and negative cases. Overall, we believe the simplicity and effectiveness of COVID-Net USPro make it a promising tool to aid the COVID-19 screening process using ultrasound images. We hope the open-source release of COVID-Net USPro helps researchers and clinical data scientists to accelerate innovations in the combat against the COVID-19 pandemic that can ultimately benefit the larger society.

Several future research directions can be explored to further improve the network. First, some additional steps in data augmentation and preparation can be taken to improve the data quality and the dataset size. In this work, ultrasound images captured with linear probes were excluded. More image augmentation and preparation techniques can be experimented with to include images captured with a linear probe and increase the data size. Second, we used simple cropping to filter out the pleura region of the images. A more procedural image segmentation step could be added to include only clinically relevant areas of the images for network construction to further improve network performance from the explainability standpoint. Lastly, we used COVIDx-US, which is a public dataset that includes data of various sources and quality. Network training could be improved by only using high-quality input ultrasound data with sufficient/specific image depth, collected, and reviewed systematically. For this purpose, a data collection protocol might be required to capture ultrasound images in a standardized manner from a set of consented participants.

## Figures and Tables

**Figure 1 sensors-23-02621-f001:**
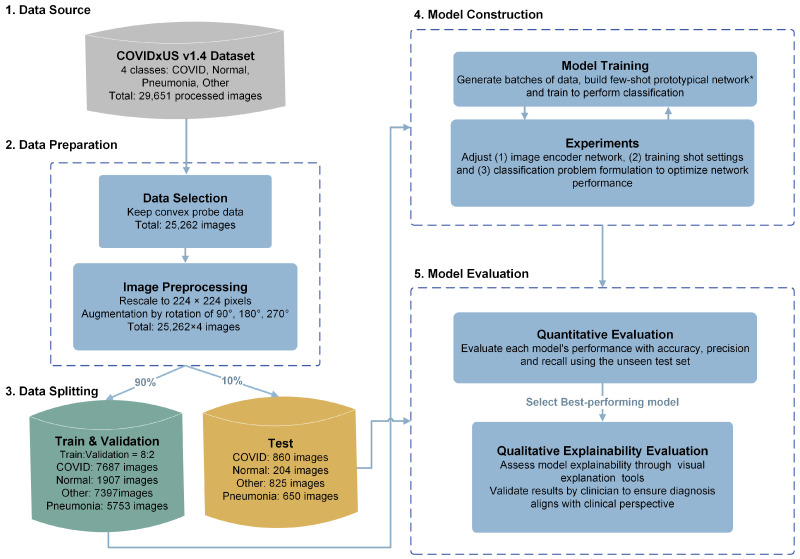
High-level conceptual flow of the analysis.

**Figure 2 sensors-23-02621-f002:**
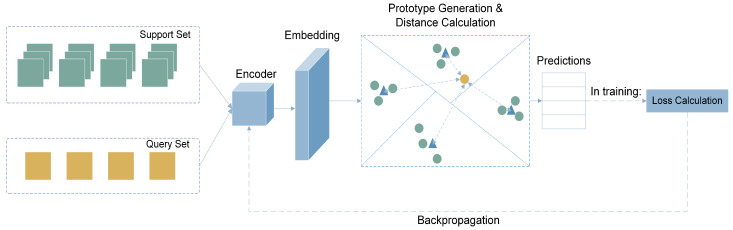
COVID-Net USPro, network architecture design.

**Figure 3 sensors-23-02621-f003:**
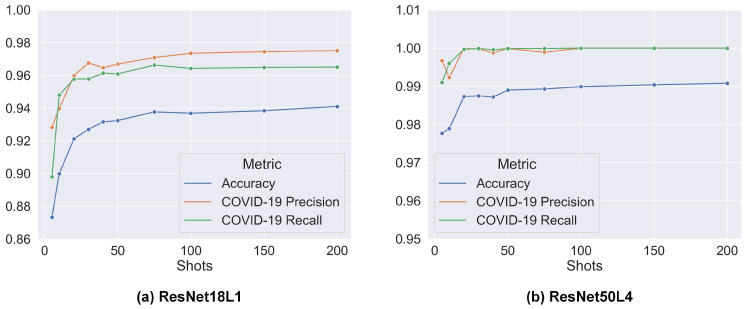
Performance results with increasing shots trained under 4-way condition: (**a**) Pre-trained ResNet18 with trainable parameters on the final connected layer and setting out features as the number of classes (ResNet18L1). (**b**) Pre-trained ResNet50 with trainable parameters on the last 3 convolutional layers and final connected layer (ResNet50L4).

**Figure 4 sensors-23-02621-f004:**
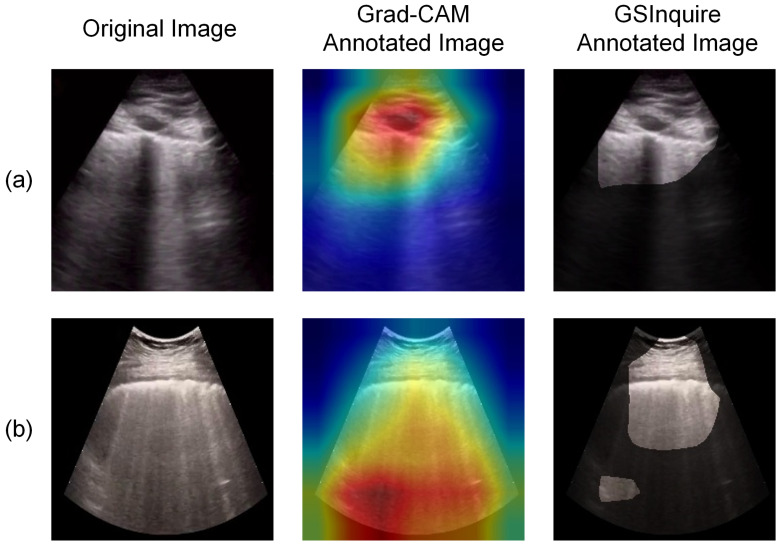
COVID-19 positive case examples correctly classified by COVID-Net USPro with high confidence: (**a**) an example of wrong decision factors. (**b**) an example of a decision made based on disease-related patterns.

**Figure 5 sensors-23-02621-f005:**
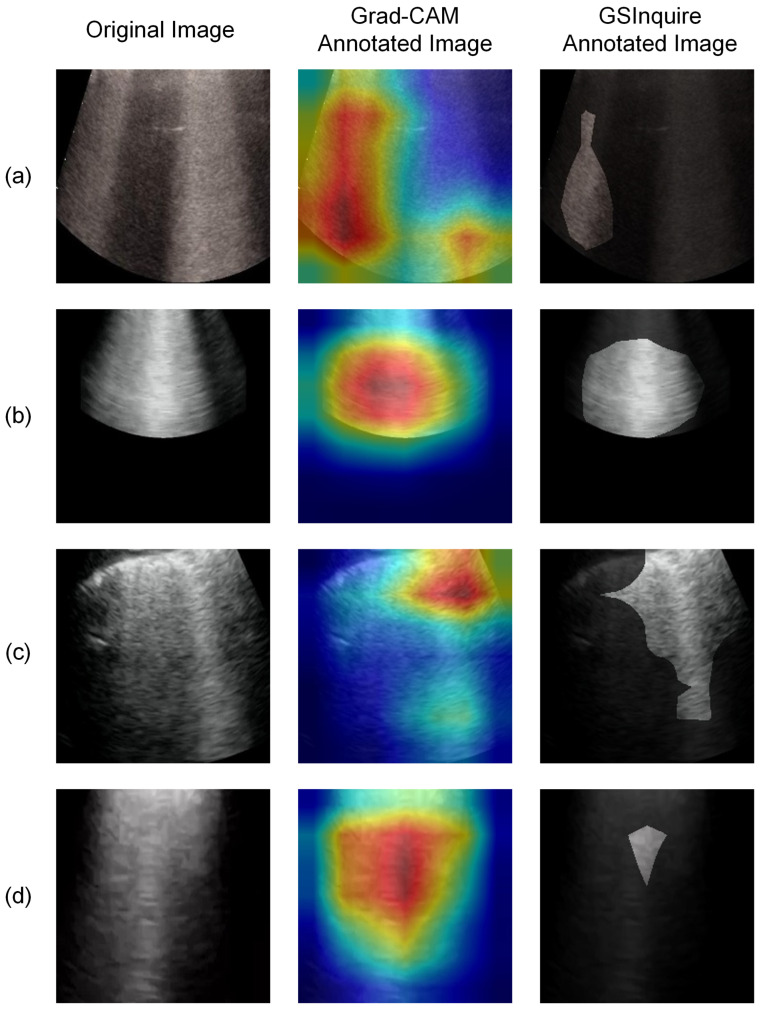
Four cropped COVID-19 positive examples predicted correctly with high confidence by COVID-Net USPro (**a**–**d**), while recognizing disease artifacts, e.g., extended B-lines.

**Table 1 sensors-23-02621-t001:** Summary of classification results for 5-shot and 100-shot conditions.

Scenario	No. Shots	Model	Accuracy	Precision	Recall
2-way	5	ResNet18L1	0.9310	0.9384	0.9380
2-way	5	ResNet18L5	**0.9955**	**0.9983**	**0.9993**
2-way	5	ResNet50L1	0.9605	0.9567	0.9740
2-way	5	ResNet50L4	0.9910	0.9933	0.9900
2-way	100	ResNet18L1	0.9741	0.9741	0.9744
2-way	100	ResNet18L5	**0.9999**	**1.0000**	0.9999
2-way	100	ResNet50L1	0.9953	0.9954	0.9953
2-way	100	ResNet50L4	**0.9999**	0.9999	**1.0000**
3-way	5	ResNet18L1	0.9487	0.9470	0.9370
3-way	5	ResNet18L5	0.9947	**0.9992**	0.9920
3-way	5	ResNet50L1	0.9690	0.9761	0.9610
3-way	5	ResNet50L4	**0.9953**	0.9940	**0.9940**
3-way	100	ResNet18L1	0.9863	0.9858	0.9857
3-way	100	ResNet18L5	**0.9998**	0.9994	0.9998
3-way	100	ResNet50L1	0.9973	0.9947	0.9980
3-way	100	ResNet50L4	**0.9998**	**0.9995**	**1.0000**
4-way	5	ResNet18L1	0.8733	0.9283	0.898
4-way	5	ResNet18L5	**0.9855**	**0.9980**	**0.9970**
4-way	5	ResNet50L1	0.9320	0.9622	0.9550
4-way	5	ResNet50L4	0.9777	0.9967	0.9910
4-way	100	ResNet18L1	0.9369	0.9735	0.9643
4-way	100	ResNet18L5	0.9884	**1.0000**	**1.0000**
4-way	100	ResNet50L1	0.9811	0.9952	0.9962
4-way	100	ResNet50L4	**0.9899**	**1.0000**	**1.0000**

## Data Availability

In this research, we used the COVIDx-US dataset (v1.4.). COVIDx-US is an open-access benchmark dataset of COVID-19 related ultrasound imaging data and is available to the general public at NRC COVIDx-US GitHub repository https://github.com/nrc-cnrc/COVID-US, accessed on 2 February 2023.

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
