# Peer review of "COVID-Net USPro: An Explainable Few-Shot Deep Prototypical Network for COVID-19 Screening Using Point-of-Care Ultrasound"

_sensors, 2023, doi:10.3390/s23052621_

Round 1

Reviewer 1 Report

This paper introduced an explainable few-shot deep prototypical network (COVID-Net USPro) that monitors and detects COVID-19 positive cases from minimal ultrasound images. The authors should consider the following issues in their paper:

1.       The title is very long. It should be rewritten in a short, concise way to represent the main content of the paper.

2.       The abstract section is inconsistent and very long. Besides, it does not reflect the main contributions of the manuscript. The authors should rewrite the abstract section to mention the paper's main purpose, primary contributions, experimental results, and global implications.

3.       The link to the code does not work.

4.       The related work should be a section, not a subsection from the introduction section. Besides, the structure of the paper should be discussed at the end of the introduction section to follow it easily.

5.       At the end of the related work section, the authors should discuss the limitations of the current related work. Then, they should discuss how they overcame these limitations in their proposed system.

6.       In the related work section, it is highly recommended to refer to the discussed studies by the first author's last name to follow the discussion easily.

7.       The paper needs intensive proofreading, containing many long, inconsistent sentences and paragraphs. Besides, the manuscript contains many grammar errors and typos.

8.       The authors should compare their model with other works from the literature review.

9.       It is highly recommended that the hyperparameters of the network be listed in a table.

Reviewer 2 Report

General comments
=============
Due to the lack of well-annotated medical image data, this paper studies the few-shot (only five image required) deep prototypical network to monitor and detect COVID-19 infection from ultrasound image. The network achieves comparable accuracy and precision with previous methods. The data is opened and the network's explainability and interpretability are invested using different visualization tools.

For the reviewers, the completeness of this paper is fair, the technical development of this paper is moderate and novelty of this paper is low.

Specific comments
=============
Major comments
---------------------
1. Page 4, line 145-146, There is no data used for the validation. Why? This validation process gives information that helps us tune the model's hyperparameters and configurations accordingly.

2. Page 5, line 188. The Sqaured Euclidean Distance was chosen to classify the query image. Any reason behind it? Did the authors try other distance metric? How are the results of other distance metric?

3. Table 1. There only summarize the result from 5-shot and 100 shot, while the author declare the experiments among 5, 10, 20, 30, 40, 50, 75, 100, 150, 200 shot. Please summary all the detailed results in the appendix. It would be important to report it.

4. Figure 3. Why the performance between ResNet18L1 and ResNet50L4 is so different? Any analysis for it?

5. Page 9, line 332. For the reviewer, manual data exclusion has the potential to manipulate data and will effect the final network performance. Why do this? Did the author quantify the difference between exclusion low quality data and inclusion low quality data?

Minor comments
---------------------

1. Page 2, line 149, data augmentation in this paper only includes rotation. Why the authors do not consider the crop, flip and image translate?

2. Page 6, line 233. The authors only report 4 best encoders using ResNet18. What is the performance from other types of decoders? 

Round 2

Reviewer 1 Report

Thanks very much to the authors for their effort in improving their manuscript. They satisfied most of my comments. Besides, I do not have more comments for them.

Reviewer 2 Report

For the reviewer, the author addresses all the comments and questions. The new version of the paper is a well-written and clear version, which is ready to be published.